# A Basic Review on Estrogen Receptor Signaling Pathways in Breast Cancer

**DOI:** 10.3390/ijms24076834

**Published:** 2023-04-06

**Authors:** Léa Clusan, François Ferrière, Gilles Flouriot, Farzad Pakdel

**Affiliations:** Université de Rennes, Inserm, EHESP, Irset (Institut de Recherche en Santé, Environnement et Travail)—UMR_S 1085, F-35000 Rennes, France

**Keywords:** estrogen receptor, breast cancer, endocrine resistance

## Abstract

Breast cancer is the most common cancer and the deadliest among women worldwide. Estrogen signaling is closely associated with hormone-dependent breast cancer (estrogen and progesterone receptor positive), which accounts for two-thirds of tumors. Hormone therapy using antiestrogens is the gold standard, but resistance to these treatments invariably occurs through various biological mechanisms, such as changes in estrogen receptor activity, mutations in the ESR1 gene, aberrant activation of the PI3K pathway or cell cycle dysregulations. All these factors have led to the development of new therapies, such as selective estrogen receptor degraders (SERDs), or combination therapies with cyclin-dependent kinases (CDK) 4/6 or PI3K inhibitors. Therefore, understanding the estrogen pathway is essential for the treatment and new drug development of hormone-dependent cancers. This mini-review summarizes current literature on the signalization, mechanisms of action and clinical implications of estrogen receptors in breast cancer.

## 1. Introduction

Among the 19.3 million new cancer cases worldwide in 2020, breast cancer was the most diagnosed cancer, just ahead of lung cancer (accounting for 11.7% and 11.4% of new cancer cases, respectively). Lung cancer continues to account for the majority of deaths among all genders (18% of the 10 million cancer deaths worldwide) while breast cancer is the most deadly cancer in women (6.9% of deaths among all genders, but 15.5% of female deaths). Although breast cancer most frequently affects women, about 1 in 100 breast cancers diagnosed in the United States is found in a man with the same characteristics as in women [1].

In healthy tissue, the mammary epithelium, whether ductal or lobular, consists of two types of cells: the luminal cells, which produce milk; and the basal cells, which have a muscular contraction activity, allowing for the mechanical triggering of milk secretion [2]. Luminal cells express hormone receptors such as estrogen (ER), progesterone (PR) or prolactin receptors, but are also characterized by the expression of cytokeratins 8 and 18 and the junctional molecule EpCam (epithelial cell adhesion molecule) [3]. Basal cells are characterized by the expression of cytokeratins 5, 14 and 17, P-cadherin, desmosomal cadherins and various markers associated with smooth muscle [3,4,5]. The development and differentiation of mammary cells are controlled mainly by estrogen, progesterone and prolactin. Growth factors such as amphiregulin (AREG), a member of the epidermal growth factor (EGF) family, insulin-like growth factor (IGF) and fibroblast growth factor (FGF) families are also essential mediators of estrogen in mammary gland development during puberty [3,6,7].

Breast cancer is a multifactorial and very heterogeneous disease that refers to the abnormal proliferation of the lobular and ductal epithelium of the breast which becomes malignant and forms a tumor. Furthermore, intratumoral heterogeneity increases the ability of cancer cells to reprogram their gene expression profile and modify their behavior to adapt to signals from the microenvironment. These characteristics contribute to the progression of malignant tumors and have a negative impact on prognosis and response to treatments. Indeed, after developing locally, the cancer can become invasive by spreading metastases to other organs such as the bones, lungs, liver or brain [8]. Thus, there are not one but many breast cancers, differing in location, histological type, molecular signature, and particularly in their dependency on hormones to proliferate. The combination of these factors gives rise to breast cancers whose aggressiveness and prognosis are very variable and therefore require different therapeutic management [9].

This mini-review aims to present an overview of estrogen signaling in breast cancer, with particular emphasis on the genomic and non-genomic action of the ER alpha subtype. The main mechanisms involved in resistance to hormone therapy in ER-positive breast cancer have also been described.

## 2. Background on Breast Cancer

### 2.1. Classification

The classification of breast cancers follows the recommendations of the WHO (World Health Organization), which are regularly revised to improve patient care [10]. Tumor classification is crucial in grading the cancer and choosing the most effective treatment to beat the disease. For example, the presence or absence of hormone receptors determines, among other factors, the need for hormone therapy or targeted therapies.

One of the main parameters for the classification of breast cancers is their molecular profile, as described in 2000 by Perou et al. [11]. This study highlighted the heterogeneity of breast cancers at the molecular level through the differential expression of a panel of key genes. The expression profiles obtained allow breast cancer to be divided into four main groups, namely luminal A (50–60% of cases), luminal B (10% of cases), human epidermal growth factor receptor 2 (HER2)-positive (20% of cases) and basal-like triple-negative tumors, which represent about 10% of breast cancers. Another subgroup has also been described as a normal-like subcategory which resembles the luminal A group but with a worse prognosis. This subgroup accounts for 10% of the luminal A group. In daily clinical practice, these tumor groups are usually identified by immunohistochemical markers such as ERα, PR and HER2. Indeed, these biomarkers that are used for tumor identification, can replace transcriptomic data to match the different types of breast cancer for prognostic and therapeutic purposes (Table 1). Luminal cancers, characterized by the expression of ERα, are the least aggressive. They may express PR, and only type B also expresses HER2. Luminal B cancers are then more aggressive than luminal A (low proliferative), due to a higher expression of genes involved in cell proliferation and cell cycle, such as Aurora kinase A and KI-67. Luminal B cancers also have lower expression of luminal-related genes, such as PR and FOXA1 [12,13,14]. In contrast to luminal A tumors, luminal B tumors are associated with a higher rate of p53 mutations (29% vs. 12%). In HER2-positive cancers, ERα and PR receptors are not expressed. Overexpression of the HER2 receptor activates different signaling pathways such as Ras/MAPK (mitogen-activated protein kinases) or PI3K/AKT (Phosphoinositide 3-kinases/ Protein kinase B), resulting in increased cell proliferation and survival. This favors the development of metastasis and makes this type of breast cancer more aggressive than luminal cancers [9]. Basal-like tumors that do not express the major biomarkers (ERα, PR and HER2) were therefore defined as triple negative in relation to the other groups. The gene expression characteristics of this breast cancer subtype include keratin 5, keratin 17, intergrin-B4, laminin, and a high expression of proliferation-related genes such as amplification of MYC, CDK6, CCNE1. In addition, triple-negative breast cancer is characterized by altered DNA repair such as deletion of BRCA2, PTEN, and MDM2; and a high mutation rate of TP53. These tumors are more aggressive and heterogeneous and have a high proliferation index, a high histological grade and a high rate of local and distant recurrence and therefore a poor prognosis [15,16].

### 2.2. Occurrence and Risk Factors

The biological mechanisms leading to the occurrence of breast cancer are complex and can be caused by a number of risk factors, summarized in Figure 1. These are related to genetics, hormonal exposure, lifestyle, food and alcohol consumption, obesity and environment [12,17,18,19,20,21]. Genetic and hereditary factors account for less than 10% of breast cancers. In this case, most patients carry a mutation in BRCA1 and BRCA2 (Breast Cancer Associated gene 1 and 2) whose function is DNA repair and cell cycle checkpoint activation. Alteration of these genes results in chromosomal instability and impairs checkpoint control, which promotes tumor development [22]. One of the most important risks of breast cancer is hormonal exposure. Long-term exposure to estrogen, both endogenous and exogenous, increases the risk of breast cancer [23]. It is also relevant to note that hormonal treatment of menopause (estrogen plus progestin) has been associated with an elevated risk of in situ and invasive breast tumors [24]. Concerning androgens, studies are contradictory and do not show an obvious relationship between circulating androgen levels and the risk of breast cancer [25]. However, in premenopausal women, McNamara et al. showed that high circulating testosterone levels appear to increase the risk [26]. In postmenopausal women, a high serum testosterone level may be also an important prognostic factor for breast cancer recurrence [27]. Environmental factors such as ionizing radiation, air pollutions, heavy metals and the chronic exposure to chemicals such as polychlorinated biphenyl, polycyclic aromatic hydrocarbons, organic solvents and organochlorine pesticides and insecticides are also highly regarded as contributing to breast tumorigenesis [28,29,30]. The risk factors are more prevalent in developed countries, where first pregnancies are later, breastfeeding is less common, hormonal treatments are more frequently used, and lifestyles are more sedentary. This explains why the incidence rate of breast cancer is higher than in developing countries, despite the fact that screening campaigns are less widespread. However, changing lifestyles in developing countries and Asia are leading to an increase in breast cancer incidence and, consequently, in breast cancer mortality. The implementation of prevention and early detection campaigns is therefore particularly important to counter the progression of this cancer, especially in developing countries [1].

The deregulation of many cellular actors leads to the uncontrolled proliferation and survival of cancer cells [31], for example, the overexpression of oncogenes such as MYC, a transcription factor that drives the expression of genes involved in cell proliferation and survival. In addition to breast cancer, MYC expression is increased in many other human cancers, both through gene transcription and protein stability [32,33]. Similarly, overexpression or activation of receptor tyrosine kinases and their signaling pathways, such as the EGFR1 (Epidermal Growth Factor Receptor 1) or HER2 family, are often observed. Activation of these factors induces numerous signaling pathways such as Ras/MAPK, PI3K/AKT or PLC/PKC, which are involved in the regulation of cell proliferation and survival (Figure 2). Within this family, the overexpression of EGFR1 and HER2 in particular plays a role in the development of breast cancer [34]. On the other hand, IGF1R (Insulin like Growth Factor 1 Receptor) is the receptor for the IGF1 protein that plays an important role in the development and functionality of the mammary gland. Binding of IGF1 to its receptor leads to activation of IGF1R by phosphorylation, which activates the PI3K/AKT and Ras/MAPK signaling pathways with proliferative and anti-apoptotic activity (Figure 2). Overactivation of these pathways can be caused by increased circulating IGF1 levels or overexpression of IGF1R, contributing to the uncontrolled proliferation and survival of breast cancer cells [35]. Finally, the loss of expression or function of tumor suppressor genes such as BRCA1 and BRCA2 may contribute to the carcinogenesis of breast tissue. These genes code for proteins involved in the repair of DNA double- strand breaks by homologous recombination. Mutation of these genes alters these DNA repair mechanisms, which increases the risk of acquiring mutations and contributes to the development of various cancers, in particular those of the breast and ovary [36]. It is also worth noting that the p53 protein encoded by the TP53 gene is a transcription factor that activates the expression of genes involved in cell cycle arrest or apoptosis in response to a variety of cellular stresses (DNA damage, hypoxia, oxidative stress, etc.). This antiproliferative activity is then essential to prevent the development of cancer cells, explaining why TP53 mutation plays an important role in many cancers including breast cancer [37]. Furthermore, the PTEN (Phosphatase and Tensin homolog) gene encodes a phosphatase that participates in the inhibition of the PI3K/AKT pathway (Figure 2). The loss of PTEN expression or activity then participates in the deregulated activation of this signaling pathway, which plays a prominent pro-proliferative and metastatic role in the development of breast cancer [38]. Also of note is the role of the androgen receptor (AR) in breast tumorigenesis. AR has been found in both ER-positive and negative breast tumors. It appears that AR can either inhibit or stimulate the growth of ER-positive breast cancer cells; however, it promotes proliferation and expansion of ER-negative cells through non-genomic events that require Src protein kinase and PI3K activities [25,39].

### 2.3. Diagnosis and Treatments

The initial diagnosis of breast cancer is based on a combination of clinical examination, imaging and biopsy analysis. The pathological examination of the biopsy systematically includes evaluation of the prognostic and predictive factors such as tumor size, histoprognostic grade Scarff-Bloom-Richardson I to III; mitotic index or Ki67; evaluation of ERα and PR status; and the search for HER2 amplification. Ductal carcinoma is the most common type of breast cancer, accounting up to 75% of patients, with lobular carcinoma (10–15%) and mixed ductal/lobular carcinoma accounting for the remaining patients. In the United States, at the time of diagnosis, more than 60% of breast tumors are located in the breast, nearly 30% have spread to regional lymph nodes and 5–10% of tumors are characterized as metastatic [40]. It is also notable that luminal A breast cancer is most frequent in white women, accounting for 67%, compared with 57% in premenopausal white women, 55% in African American women, and 40% in premenopausal African American women [41]. After determining the type of breast cancer and its stage of development, different treatment options exist and are combined according to each patient's profile.

The treatment of breast cancer is generally based on (i) surgery which consists of the removal of the tumor while preserving the mammary gland as much as possible (75% of cases), or total removal of the mammary gland, depending on the extent of the cancer; (ii) radiotherapy which is a targeted irradiation aimed at destroying any residual tumor cells following surgical treatment; (iii) chemotherapy such as anthracycline and taxane which are commonly used in HER2 amplified and triple-negative tumors, and represents a non-targeted treatment that destroys dividing cells and can be given before surgery to reduce the size of the tumor and/or after surgery if there is a high risk of relapse [40,41,42]; (iv) hormone therapy, which is a specific treatment for luminal breast cancer and is based primarily on the application of estrogen receptor antagonists in pre- and postmenopausal women or decreasing estrogen level in exclusively postmenopausal women; (v) other specific treatments, such as targeting HER2 with the monoclonal antibody trastuzumab (Herceptin), or inhibiting the PI3K-AKT-mTOR pathway, which is involved in cell proliferation and survival and plays a key role in hormonal escape [43]. Neoadjuvant endocrine therapy is also becoming increasingly important. Of note, the combination of CDK4/6 inhibitors, acting at the cell cycle level, and endocrine therapy has been shown to be beneficial in patients with metastatic ER+/HER2- [44,45]. Thus, all these treatments significantly extend the survival of patients [42,43,44,45,46].

Follow-up of patients (usually by imaging) is essential to ensure the success of the chosen treatment, to adjust the therapeutic management, and to verify the absence of relapse. In order to select the most appropriate treatment for each patient according to the evolution of her cancer, another strategy developed in recent years is the analysis of circulating tumor DNA. This non-invasive method allows for the detection of oncogenic mutations, the presence of which can guide the therapeutic management towards the choice of specific treatments [47]. Indeed, PIK3CA mutation was found in 83% of the metastatic tumor tissue [48]. Differentially methylated regions at the genome level between normal breast tissue and triple-negative breast cancer tissue demonstrate the utility of methylation signatures as non-invasive liquid biopsy markers for breast cancer diagnosis [49]. Circulating tumor cells can also be detected in the blood and used as a biomarker for the efficacy or otherwise of treatment of metastatic cancer [50]. However, the value of using these approaches needs to be validated by additional clinical trials before it can be applied routinely.

## 3. ER-Positive Breast Cancer

Approximately 70% of patients have hormone-dependent breast cancer with tumor cells, called luminal A and B, expressing the ER. In these tumors, estrogens are the principal signals that play a major role in tumor cell growth and progression. The cellular action of estrogens is primarily mediated by the nuclear ERα, ERβ and the membrane G protein-coupled ER (GPER, called also GPR30). The ERα, being considered to be the receptor most involved in the development of breast cancer [8], therefore constitutes therefore a pivotal target for breast cancer therapy.

### 3.1. Molecular Mechanism of ER 

#### 3.1.1. Genomic Action

ERα is a transcription factor regulating the expression of genes involved in cell cycles, proliferation and apoptosis. Indeed, activation of ERα allows the expression of factors such as MYC, Cyclin D1, FOXM1, GREB1, BCL2 or amphiregulin, IGF-1 and CXCL12, which have oncogenic potential, increasing cancer cells proliferation and the risk of DNA damage in response to the estrogens [9]. Once estrogen (E2) is bound to ERα, it allows the receptor to change its conformation in an active form, dimerize, translocate to the nucleus and interact with transcriptional coactivators (Figure 3). It is of note that unlike E2, antagonistic molecules such as tamoxifen induce inactive conformation of ERα, which recruit transcriptional corepressors [51]. Ligand-activated ERα then binds to estrogen-responsive elements (EREs) within the promoters of target genes. ERα can also interact with transcription factors, such as activator protein 1 (AP1) and specific protein 1 (SP1) via serum responsive elements (SREs) to regulate genes lacking ERE in their promoters (Figure 3). This genomic action thus regulates the transcription of hundreds of target genes involved in cell growth and differentiation [52,53,54]. The deregulation of ERα expression, activity or its coregulators and target genes then plays a prominent role in the development of the majority of breast cancers, known as ERα+ or luminal. It should be noted that the two other receptors, ERβ and GPER, can be stimulated by estrogens. ERβ is a nuclear receptor homologous to ERα, encoded by the ESR2 gene and with a structure similar to ERα [55]. Like ERα, ERβ is expressed in many reproductive organs such as mammary epithelial cells, ovaries, uterus and testes, as well as non-reproductive organs, such as lungs, adrenal or adipose tissue or brain. This receptor interacts with some ERα transcriptional coregulators and shares ligands similar to ERα, such as estrogens and SERMs (selective estrogen receptor modulators), but with different affinity. In contrast to ERα, activation of ERβ generally results in inhibition of proliferation and induction of apoptosis, but these effects depend on the tissue studied, the cell context, transcriptional coactivators, and whether ERα is coexpressed [55]. ERβ expression in breast tumor cells then tends to correlate with a favorable prognosis, but some studies indicate the opposite. Indeed, previous studies indicate that ERβ and its isoforms as well as certain coactivators such as AIB1, NF-kB and TIF-2 tend to coregulate breast cancer cell proliferation and progression [56,57]. These specific coactivators are associated with poor clinical outcomes and were correlated with high ERβ expression in high-grade breast tumor subtypes, suggesting that they may mediate ERβ proliferation in breast cancer cells [58]. Thus, further work remains necessary to better understand the physiological role of this receptor and its involvement in breast carcinoma [59,60].

Both ligand-dependent and -independent activation of ERα have been reported [61,62]. ERα is the target of numerous post-translational modifications that are crucial for the regulation of ligand- independent ERα and may affect its stability, dimerization, subcellular localization, DNA binding or interaction with cofactors [63]. In particular, phosphorylation of ERα following activation of certain intracellular kinases by growth factors is a key mechanism of ligand-independent activation of ERα [64,65,66]. Notably, phosphorylation of serine residues 118 (S118), S167, S305 and tyrosine 537 (Tyr537) increases ERα activity through interactions with coactivators in breast cancer cells [66,67,68,69,70,71]. ERα is also acetylated at different lysine residues by the acetylase p300. Interestingly, acetylation of lysine 266/268 stimulates ER transcriptional activity, while acetylation of lysine 302/303 inhibits ERα activity [72].

#### 3.1.2. Nongenomic Action

The interplay between growth factors such as EGF or IGF and sex steroid (estrogen or androgen) signaling has been previously analyzed at different levels [73,74]. Indeed, in addition to its translocation into the nucleus and its function as a genomic pathway transcription factor, a small proportion of cytosolic or membrane-anchored ERα also rapidly and transiently exerts non-genomic activity [75,76]. This involves the rapid activation of intracellular signaling pathways including cAMP (cyclic adenosine monophosphate) production or the activation of growth factor receptors, PI3K/AKT or Ras/MAPK pathways (Figure 3). Upon E2 binding, ERα rapidly forms a cytoplasmic complex with several proteins including the Src protein kinase and the p85 subunit of PI3K, triggering the MAPK and Akt pathways after only 2 to 15 min of ligand binding [77,78,79]. The mechanisms of ERα addressing the plasma membrane and its non-genomic activity are based on post-translational modifications. For example, palmitoylation of cysteine 447 of human ERα is essential. The binding of palmitate to this residue increases the hydrophobicity of the receptor and its anchoring to the caveolae which are regions of the plasma membrane enriched in cholesterol [80]. Interestingly, this association of ERα with caveolin-1 in the plasma membrane triggers non-genomic signaling pathways, activation of cyclin D1 and cell proliferation [80,81,82]. Furthermore, Le romancer et al. [83] demonstrated that methylation of ERα on arginine 260 by the arginine methyltransferase PRMT1 is required for the interaction of ERα with Src and p85 partners and thus the stabilization of the estrogen-induced ERα/Src/p85 complex. in addition, Src and PI3K activity is essential for ERα methylation and thus for ERα/Src/p85 association and downstream Akt activation [63,84,85]. It is also noteworthy that ERα methylation occurs only 5 to 15 min after ligand stimulation [63]. Note also that IGF-1 causes rapid methylation of ERα by PRMT1 and triggers ERα binding to the IGF-1 receptor in MCF-7 breast cancer cells. Interestingly, IGF-1 receptor expression was found to be positively correlated with ERα/Src and ERα/PI3K interaction in a cohort of breast tumors [84,85]. On the other hand, phosphorylation of Tyr537 of ERα plays a key role in the interaction of ERα with Src kinase, activation of the MAPK pathway and cell proliferation [86].

GPER, a seven transmembrane domain protein, that can trigger rapid cellular effects of estrogen [87], is also expressed in different organs such as liver and adipose tissue in addition to the mammary gland [88]. GPER expression has also been reported in several types of breast cancer cells [89]. Its stimulation by estrogen increases cAMP concentrations and the mobilization of intracellular calcium [90,91]. This also results in the activation of signaling cascades such as PI3K/AKT and Ras/MAPK, ultimately regulating the transcription of genes for cell proliferation and survival [92,93]. These genes include c-fos, cyclin D, B and A which are involved in the cell cycle and promote cell proliferation. They are induced by GPER while other genes such as BAX, caspase 3 and BCL2, involved in the process of cell apoptosis, are downregulated by GPER [94,95,96,97]. In a mouse xenograft model of breast cancer, GPER activation increases tumor growth and expression of HIF1, VEGF, and the endothelial marker CD34 [98]. Conversely, in this model, the use of a GPER inhibiting peptide exerts apoptosis and induces a significant decrease in the size of triple-negative mammary tumors in mice [99]. Activation of the membrane receptor GPER would then appear to have a pro-tumor effect and could play a role in the acquisition of resistance to hormone therapy, with SERMs acting as agonists of this receptor [100,101,102]. In fact, tamoxifen exerts an agonist effect by inducing gene expressions involved in breast tumorigenesis [100,103]. Moreover, in ERα-positive patients receiving tamoxifen treatment, the presence of GPER strongly correlated with worse relapse-free survival [96,98]. However, as for ERβ, further work is needed to better identify the functions of this receptor and consider it as a therapeutic target in breast cancer [104].

The nuclear and membrane-initiated pathways of ERα should not be completely dissociated. Indeed, there is a possibility of dialogue between genomic and non-genomic actions. One way in which these two major ERα pathways are connected is through phosphorylation of ERα or its cofactors [105,106,107]. For example, E2-mediated transcriptional activation of cyclin D1 via the AP-1 binding site requires MAPK activity and formation of an ERα/Src/PI3K complex [106,107], indicating that there is a convergence of genomic and non-genomic signaling on E2 target genes [108]. However, previous studies have suggested that the non-genomic action of ERα may be associated with resistance to endocrine therapy and poor prognosis in breast cancer [84,109,110]. We also showed that during cancer progression, a transition to the monomeric state of ERα induces a decrease in the genomic activity of ERα and promotes its non-genomic activity [111].

### 3.2. ERα Variants and Mutations

In addition to the full-length ERα which represents a 66 kDa protein, two major variants of lower molecular weight, ERα 46 kDa and ERα 36 kDa, have been identified [112,113,114,115,116,117]. These variants which originate from alternative splicing, have been detected in healthy and cancerous breast tissues, as well as in various breast cancer cell lines [116,118,119]. The ERα46 isoform differs from the full-length ERα66 only in the absence of the N-terminal activation function 1 (AF-1), whereas the ERα36 isoform, lacks both transactivation domains, AF-1 and AF-2, but conserves the DNA-binding domain and a part of ligand-binding domains [112,116]. The overall structure of ERα36 is identical to ERα46 except for a unique sequence of 27 amino acids in place of the last 138 amino acids in the C-terminus of ERα46 and ERα66. The two truncated isoforms, ERα46 and ERα36, show partially different activities than the classical ERα66, notably for ligand binding, transactivation, interaction with coregulators and subcellular localization [113,117,120]. In particular, ERα36 is mainly located in the cytoplasm and at the plasma membrane and expressed in both ER-positive and ER-negative breast cancer cells [112,116,120]. This receptor is also able to bind to DNA and thus inhibit the genomic activity of ERα66. In the cytoplasm, it exerts a non-genomic activity in the presence of estrogens, allowing for the activation of multiple pathways, notably the MAPK pathway, which contribute to cancer aggressiveness and progression. ERα36 can also mediate the agonistic effects of tamoxifen and induce resistance to antiestrogens [121,122,123,124,125]. However, further research is needed to fully identify the biological activities of these ERα variants and to determine whether they constitute a potential therapeutic target. Also of note is that evaluation of ER status in breast tumors is important to provide information on diagnosis and treatment choice. However, current methods using immunohistochemistry (IHC) analysis have limitations and do not discriminate these ERα subtypes in the biopsies. Accurate characterization of these variants therefore requires specific antibodies for each of these isoforms.

Previous studies have shown that mutations on the ERα gene occur in less than 5% in primary tumors while they are significantly increased up to 50% in the metastatic-resistance tumors, especially in patients treated with aromatase inhibitors (AIs) [126]. So far 62 mutations have been described for the ERα gene in tumor samples. Most of these mutations (47 out of 62) occur in the ERα ligand-binding- domain (LBD), and some of them make the receptor constitutively active [127]. This suggests that these activating mutations of the ERα are only generated under the selective pressure of treatment with AIs and thus estrogen deprivation. Y537S and D538G are the two most common ERα mutations found in tumors. These mutations induce the receptor in its active conformation in the absence of ligand, enhance coactivator interactions with the receptor and strongly decrease its affinity for antiestrogens, tamoxifen and fulvestrant, indicating the importance of these mutations in the resistance to endocrine therapies. Also of note is a recent study on breast cancer cells, which showed that these ERα mutations induce epithelial-mesenchymal transition and promote the proliferation and invasion of cancer cells, in vitro and in vivo [128].

### 3.3. Hormone Therapy and Resistance

Because two-thirds of breast cancer cell proliferation is dependent on estrogen activation of ERα, the treatment of choice is hormone therapy. This consists of depriving the tumor of estrogen or blocking the activity of ERα, thus preventing breast cancer recurrence and increasing the overall survival of patients. Of the ERα antagonists, tamoxifen, which belongs to the SERM family, is the best characterized and most clinically used [129]. Unlike estrogen, tamoxifen binding to ERα induces corepressor recruitment which promotes the inhibition of breast cancer cell growth [130]. A 5-year course of tamoxifen is the treatment of choice for premenopausal and postmenopausal women. Tamoxifen has long been the only treatment for more than 20 years for advanced breast cancer. However, in the late 1990s, the development of AIs significantly transformed first and second-line treatments. These active substances (letrozole, anastrozole or exemestane) with therapeutic effect inhibit the activity of the enzyme aromatase, which allows the conversion of androgens into estrogens [131]. Indeed, while in premenopausal women most estrogen is produced by ovaries, in postmenopausal women aromatase, present in other tissues than the ovaries such as adipocyte and breast tissues, can lead to estrogen production which can favor tumor progression [131]. In the early 2000s, a new family of therapeutically active substances was developed with fulvestrant, an ERα antagonist. This family includes SERDs (selective estrogen receptor degraders), which bind to ERα, causing its inactivation by degradation [132], in contrast to SERMs, which are responsible for conformational modification of ERα by promoting the recruitment of co-repressors [129]. The binding affinity of fulvestrant to the ERα is equivalent to that of E2 without agonist activity. The use of fulvestrant is generally reserved for luminal metastatic breast cancer, or for patients who have become resistant to a first hormone therapy with IAs or SERMs [133]. Fulvestrant is used by intramuscular injection in postmenopausal women with then advanced tumors. However, new SERDs that can be prescribed orally are currently being developed and tested in clinical trials [134].

In approximately 30% of ERα+ breast cancers, breast tumor cells may escape hormonal control, thereby acquiring the ability to proliferate in the absence of estrogen stimulation. Hormonal escape is typically accompanied by the loss of the epithelial phenotype and the gain of a more migratory and invasive capacity [135]. Hormone therapy is then no longer effective in countering cancerous growth. This resistance is rarely due to a loss of ERα expression (only 10% of cases), but rather to a deregulation of the activation of this receptor, which can be stimulated independently of estrogen binding. Different mechanisms can lead to this ligand-independent activity, the most frequent alterations are (i) the acquisition of mutations rendering ERα constitutively active; (ii) epigenomic and post-translational changes in the ERα; (iii) the activation of ERα by oncogenic intracellular signaling pathways such as PI3K/AKT, and Ras/MAPK as well as growth factors (EGFR, HER2, IGF1R, FGFR), which are themselves deregulated in cancer cells; (iv) an increase in the interaction of ERα with coactivators, at the expense of its corepressors [135]. Indeed, high expression of coactivators such as AIB1 or low expression of corepressors such as NCOR1 is associated with poor clinical outcomes and is predictive of an unfavorable response to tamoxifen [58,61].

For patients who have become resistant, one therapeutic approach is to target the signaling pathways involved in ligand-independent activation of ERα. For example, aberrations of the PI3K/AKT pathway are common in ERα-positive breast cancer and could play a crucial role in tumor resistance. It is thus possible to combine hormone therapy with inhibitors of the PI3K/AKT pathway such as alpelisib and everolimus [43]. It is also possible to combine hormone therapy with inhibitors of CDK4/6 such as palbociclib, ribociclib, or of histone deacetylase (HDAC) inhibitors such as entinostat, vorinostat [44,136,137,138,139]. Indeed, overexpression of cyclin D1, present in 50% of breast cancers, leads to CDK4/6 and ERα activations and cell cycle progression [140]. Consequently, CDK4/6 inhibitors in combination with hormone therapies have become standard treatment choices for ERα-positive metastatic breast cancer [141]. Also of note that the most common epigenetic changes during cancer progression are likely DNA methylation and histone acetylation. DNA demethylating compounds such as decitabine and 5-azacytidine and HDAC inhibitors may be able to re-sensitize endocrine-resistant ER+ breast cancer [142,143]. In addition, the HDAC inhibitor, entinostat, also shows immunomodulatory action by enhancing immunocompetent monocytes, which leads to increased overall survival [144]. Furthermore, clinical trials are underway to test the efficacy of different therapeutic combinations depending on the patient profile [145]. Finally, it should be noted that ER and other steroid receptors are expressed in stem cells and may control the behavior of cancer stem cells that are, in part, responsible for treatment resistance and tumor recurrence [146].

## 4. Conclusions

Breast cancer is a heterogeneous disease that represents the leading cause of cancer-related morbidity in women [1,2,3]. Genetic and environmental factors are known to play a role in the development and progression of breast cancer, which can affect up to one in eight women. Tumors could have distinct metabolic, genetic and epigenetic profiles [147,148,149,150]. However, most breast tumors express ERα. Given the key role of this receptor in the growth and survival of hormone-dependent cancer cells, ERα constitutes an important diagnostic and therapeutic target. ERα thus regulates the transcription of many genes involved in cell proliferation through genomic and nongenomic actions [151,152,153,154,155]. While the classical genomic effects are better established, the impact of the non-genomic plasma membrane receptor pathway as well as the rapid stimulation of intracellular signaling cascades are still under active investigation. In addition, a recent study demonstrated a novel function for ERα that acts as an RNA-binding protein to enhance the survival and fitness of cancer cells by allowing them to respond to environmental stresses and nutritional conditions [156]. Attenuation of ERα activity by hormone therapy is the primary treatment for ER+ breast cancer. However, some ERα mutations or deregulation of regulatory mechanisms that ensure ERα functions as well as epigenetic factors could modify ERα action in the absence of estrogen. Particularly, these could change the dynamic nature of ERα interaction with intracellular kinases, transcription factors and DNA [127,135,150,157]. A therapeutic strategy would potentially be to target molecular and cellular mechanisms that modulate ERα functions in ERα+ breast cancer, especially for endocrine-resistant breast cancers. In view of the complexity of the resistance phenomenon, new combination therapy strategies, new molecules as well as epidrugs that target epigenetic mechanisms should be investigated. Thus, understanding the underlying mechanism of ER actions in breast cancer progression will allow for the development of new treatment strategies.

## Figures and Tables

**Figure 1 ijms-24-06834-f001:**
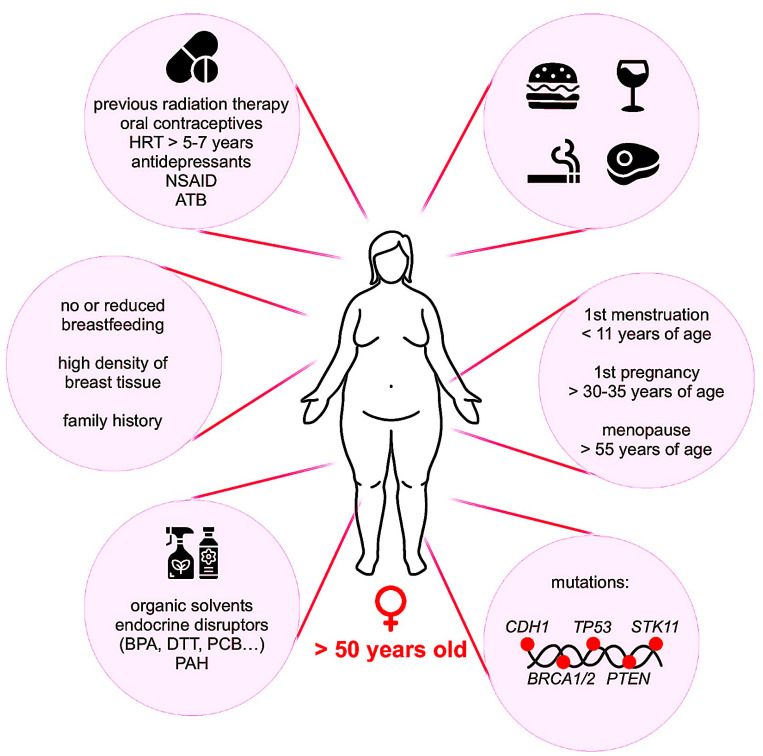
Risk factors of breast cancer. Genetic as well as environmental factors participate in breast cancer development. ATB: antibiotics; BPA: bisphenol A; DTT: dichlorodiphenyltrichloroethane; HRT: hormonal replacement therapy; NSAID: non-steroidal anti-inflammatory drugs; PAH: polycyclic aromatic hydrocarbons; PCB: polychlorinated biphenyls.

**Figure 2 ijms-24-06834-f002:**
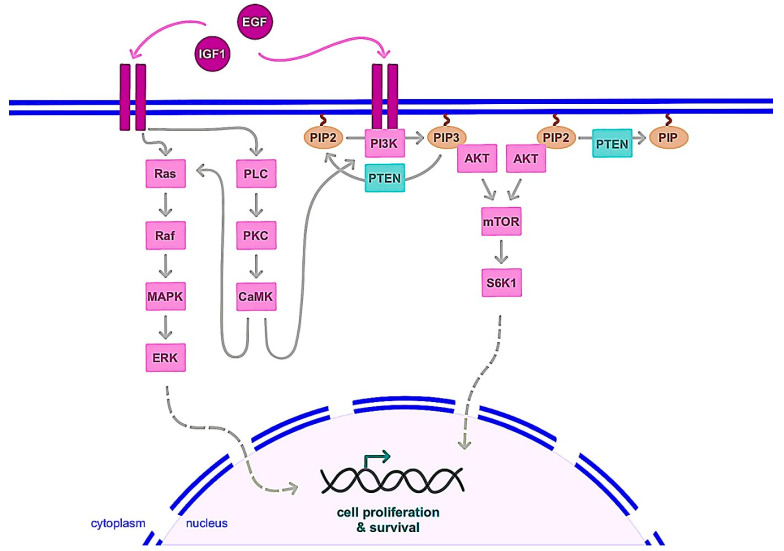
Main signaling pathways leading to cell proliferation and survival following growth factor receptors stimulation. Growth factor receptors stimulation by EGF or IGF1 for example activates signaling pathways involving notably Ras/MAPK/ERK or PI3K/AKT/mTOR, leading to transcription factors activation for the regulation of cell fate. The activation of AKT by PI3K involves PIP2 phosphorylation, which is inhibited by PTEN. CaMK: Ca^2+^/calmodulin-dependent protein kinase; EGF: epidermal growth factor; IGF1: insulin like growth factor 1; MAPK: mitogen-activated protein kinase; mTOR: mammalian target of rapamycin; PI3K: phosphoinositide 3-kinase; PIP: phosphatidylinositol phosphate; PKC: protein kinase C; PLC: phospholipase C; PTEN: phosphatase and tensin homolog; S6K1: S6 kinase 1.

**Figure 3 ijms-24-06834-f003:**
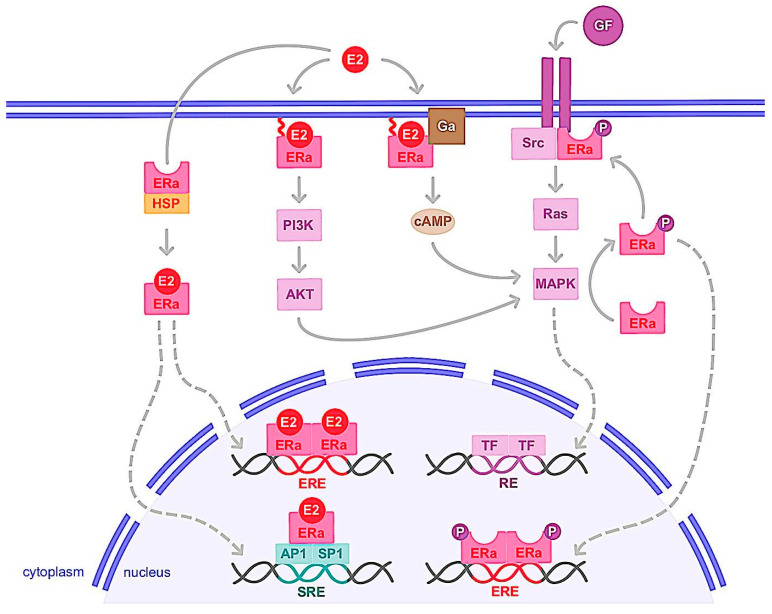
Genomic and non-genomic activities of ERα. ERα activation upon estrogen binding (E2) or after its phosphorylation (P) by cellular kinases following growth factor (GF) receptor stimulation leads to its translocation into the nucleus. There, ERα homodimer binds DNA either directly or indirectly, through estrogen responsive elements (EREs), or upon binding to other transcription factors such as AP1 or SP1 that bind DNA through serum responsive elements (SREs). This is the so-called genomic action of ERα which leads to the regulation of the transcription of target genes. ERα can also be anchored to the membrane and interacting with G proteins (Ga) or GF receptors. ERα activation is then implicated in second messenger production (cyclic adenosine monophosphate, cAMP) and signaling pathways stimulation involving PI3K/AKT or Ras/MAPK for example. This nongenomic activity of ERα eventually leads to transcription factors (TFs) activation involved in the regulation of cell proliferation and survival. cAMP: cyclic adenosine monophosphate; E2: estrogen; ERα: estrogen receptor alpha; ERE: estrogen responsive element; Ga: G alpha protein; GF: growth factor; HSP: heat shock protein; MAPK: mitogen-activated protein kinase; P: phosphate; PI3K: phosphoinositide 3-kinase; SRE: serum responsive element.

**Table 1 ijms-24-06834-t001:** Properties of the main breast cancer subtypes. Breast cancers can be classified according to the expression of certain receptors. These molecular profiles are correlated with the proliferation level of cancer cells and prognosis. In most cases, targeted therapies are available as adjuvants to surgery. ERα: estrogen receptor alpha; HER2: human epidermal growth factor receptor 2; PR: progesterone receptor.

	Luminal A	Luminal B	HER2 Positive	Triple Negative
Proportion of cases	60%	10%	20%	10%
ERα expression	++	+	-	-
PR expression	++	+	-	-
HER2 expression	-	+/-	+	-
Proliferation (Ki67)	Low	High	High	High
Prognosis	Good	Intermediate	Intermediate	Poor
Therapy	Endocrine therapy	Endocrine therapy	Anti-HER2 therapy	Chemotherapy

## Data Availability

Not applicable.

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
