# Peer review of "A Basic Review on Estrogen Receptor Signaling Pathways in Breast Cancer"

_ijms, 2023, doi:10.3390/ijms24076834_

Round 1

Reviewer 1 Report

The present manuscript aims to dissect the role of estrogen receptor in breast cancer.

There are several concerns that should be addressed.

The review is too general. Some sentences are elusive and didn’t described deeply the different aspects related to the topic of the review. It is not really clear the contribution of the manuscript to the scenario of current available data on ER and BC. There are a lot of literature data that have been not cited.

Epidemiological data are on 2020.

The introduction should be improved analyzing the different aspects described in the paragraphs.

Add the aim of the review at the end of introduction section.

Although the topic is on ER in BC, some paragraphs analyze in general several aspects of BC.

Among the several gene displaying a pivotal role in BC, AR should be mentioned considering its pivotal role particularly in triple negative BC. As an example, see doi: 10.3389/fendo.2018.00492.

The term “development” seems not really pertinent.

Risk factors should be mentioned before the molecular aspects of BC onset. Moreover, obesity, for example, has not mentioned.

Liquid biopsy represents a forefront method in cancer management. However, it has been only described the basis of the method without citing some markers useful for BC.

The diagnostic algorithm applied in BC has not been adequately described.

Steroid receptors control also cancer stem cells behavior that are responsible among the others, of resistance to therapy. BC stem cells should be described. As an example, see doi: 10.4252/wjsc.v11.i9.594

Although the title of the manuscript suggests the description of estrogen receptors in BC, it has been described prevalently ERa. Maybe, it could be more pertinent specify ERa in the title.

I suggest splitting the paragraph on “molecular mechanism of Era” in two ones on genomic and non genomic pathways.

“Conclusion section should be improved. It seems a summary of the review.

Minor comments:

I suggest English revision

I suggest a graphical abstract

I suggest improving the quality of the images

Author Response

We thank the academic editor and reviewers for highlighting the interest of the manuscript and for bringing to our attention elements to improve the former version.

The answers to the academic editor and reviewer's remarks are listed below:

Academic Editor:

The manuscript submitted by Clusan and Coworkers deals with a subject appropriate for the special issue. It seems suitable for further peer review. Nevertheless, the manuscript should be more accurately revised by the Authors, especially in the sections related to the mechanism of action of estrogen receptor in BC. Many findings in literature concerning the genomic and non genomic role of ER are almost neglected. These aspects should be extensively revised.

I would like to remember that ER is able to directly interact with the Src tyrosine kinase and PI3-K in BC cells and BC specimens. Such interaction does not require the presence or participation of GF receptors, such as EGFR or IGFR, and it has been extensively investigated using different approaches by different groups.

Again, a lot of findings have reported a key role for the G-protein-coupled estrogen receptor-1 (GPER) in BC. These findings, together with the putative intersection between genomic and non genomic events mediated by ER in BC, should be extensively discussed by the Authors, as they have opened new avenues in diagnostic, prognostic and therapeutic guidance in BC.

Authors: Thank you for these fair comments. As also suggested by review 1, we have thoroughly revised the review. We added different aspects of the genomic and non-genomic action of ER. Its interaction with Src tyrosine kinase and PI3K as well as the role of GPER in breast cancer. We also described the post-translational modifications of ERα and how these events influence both ligand-independent and non-genomic activity of ERα. We also mentioned the interactions between the nuclear and membrane-initiated pathways of ERα. Several paragraphs highlighted in yellow were included in the text at page 6 (paragraph 1 to 3) and page 7 (1st and 2nd paragraphs).

Reviewer #1:

The present manuscript aims to dissect the role of estrogen receptor in breast cancer.

There are several concerns that should be addressed.

The introduction should be improved analyzing the different aspects described in the paragraphs.

Add the aim of the review at the end of introduction section.

Authors: As suggested by the reviewer, we improved the introduction and added the objective of the study at the end of the introduction, page 1 and 2 (highlighted in yellow).

Among the several gene displaying a pivotal role in BC, AR should be mentioned considering its pivotal role particularly in triple negative BC. As an example, see doi: 10.3389/fendo.2018.00492.

Authors: We have discussed the importance of circulating androgen levels and the risk of breast cancer (1st paragraph, page 3) and the critical role of AR in BC (1st paragraph, page 4). The reference (doi: 10.3389/fendo.2018.00492) has been included in the text at these paragraphs.

The term “development” seems not really pertinent.

Authors: We have changed it to “occurrence”

Risk factors should be mentioned before the molecular aspects of BC onset. Moreover, obesity, for example, has not mentioned.

Authors: As suggested, the risk factors were mentioned before the molecular aspects of BC occurrence and obesity, and others risk factors, have mentioned now (1st paragraph, page 3).

Liquid biopsy represents a forefront method in cancer management. However, it has been only described the basis of the method without citing some markers useful for BC.

Authors: This is done (1st paragraph, page 5)

Steroid receptors control also cancer stem cells behavior that are responsible among the others, of resistance to therapy. BC stem cells should be described. As an example, see doi: 10.4252/wjsc.v11.i9.594

Authors: This is done (2nd paragraph, page 9) and the reference (doi: 10.3389/fendo.2018.00492) has been included in the text.

Although the title of the manuscript suggests the description of estrogen receptors in BC, it has been described prevalently ERa. Maybe, it could be more pertinent specify ERa in the title.

Authors: As we have now briefly described the role of ERb and GPER in breast cancer, it is preferable to keep estrogen receptor in the title. But at the end of the introduction, we have made it clear that the review is focused on the ER alpha subtype.

I suggest splitting the paragraph on “molecular mechanism of Era” in two ones on genomic and non genomic pathways.

Authors: As suggested, we have divided the genomic and non-genomic actions in two paragraphs and further described these two aspects at page 6 (paragraph 1 to 3) and page 7 (1st and 2nd paragraphs).

“Conclusion section should be improved. It seems a summary of the review.

Authors: This is done (3rd paragraph, page 9)

Minor comments:

I suggest English revision

I suggest a graphical abstract

I suggest improving the quality of the images

Authors: English has been revised. The quality of the figures has been improved. For the graphical abstract, we suggest to use also Fig. 1 as graphical abstract.

Reviewer 2 Report

Breast cancer is a common cancer which has good chances of recovery when detected at early stage. Though, due to the pure number of affected patients which may be also diagnosed at later stage there are still many deaths. The manuscript gives insight in the role of estrogen signaling in breast cancer.

At the present state, the manuscript contains some shortcomings which have to be solved before acceptance for publication

Major

1.       Introduction First paragraph:

Please mention how rare breast cancer diagnoses are in men and not just count all gender deaths

2.       Page 2, line 5-6

There are no absolute numbers in group distribution of breast cancer patients, indeed the proportion of Luminal A can be up to 75%; interestingly in some African Ccountries the basal like subtype is quite common; please give a range for each breast cancer subgroup

3.       “Luminal B cancers are then more aggressive than luminal A (low proliferative), due to a higher expression of genes involved in cell proliferation”

Which genes are differentially expressed in Luminal B compared to Luminal A contributing to proliferation?

4.       “Finally, the loss of expression or function of tumor suppressor genes such as BRCA1/2 …”

Please also mention that these mutations can be inherited and lead to early onset of related cancers in affected families

5.       Treatment of breast cancer: Please also mention neoadjuvant endocrine  treatment which is getting more important

Minor

1.       Page 2

HER2-positive, please give also the full spelling at first mentioning (also for other gene names)

Author Response

Reviewer 2:

Breast cancer is a common cancer which has good chances of recovery when detected at early stage. Though, due to the pure number of affected patients which may be also diagnosed at later stage there are still many deaths. The manuscript gives insight in the role of estrogen signaling in breast cancer.

At the present state, the manuscript contains some shortcomings which have to be solved before acceptance for publication

Major

  1. Introduction First paragraph:

Please mention how rare breast cancer diagnoses are in men and not just count all gender deaths

Authors: This is done (2nd paragraph, page 1)

  1. Page 2, line 5-6

There are no absolute numbers in group distribution of breast cancer patients, indeed the proportion of Luminal A can be up to 75%; interestingly in some African Ccountries the basal like subtype is quite common; please give a range for each breast cancer subgroup

Authors: This is done (2nd paragraph, page 4)

  1. “Luminal B cancers are then more aggressive than luminal A (low proliferative), due to a higher expression of genes involved in cell proliferation”

Which genes are differentially expressed in Luminal B compared to Luminal A contributing to proliferation?

Authors: Thank you for these fair comments. As suggested by the reviewer the following paragraph was included in the text (3rd paragraph, page 2)

Luminal B cancers are then more aggressive than luminal A (low proliferative), due to a higher expression of genes involved in cell proliferation and cell cycle, such as Aurora kinase A and KI-67. Luminal B cancers have also lower expression of luminal-related genes, such as PR and FOXA1 [12–14]. In contrast to luminal A tumors, luminal B tumors are also associated with a higher rate of p53 mutations (29% vs. 12%).

  1. “Finally, the loss of expression or function of tumor suppressor genes such as BRCA1/2 …” Please also mention that these mutations can be inherited and lead to early onset of related cancers in affected families

Authors: This is done (1st paragraph, page 3)

  1. Treatment of breast cancer: Please also mention neoadjuvant endocrine treatment which is getting more important

Authors: This is done (3rd paragraph , page 4)

Minor

  1. Page 2

HER2-positive, please give also the full spelling at first mentioning (also for other gene names)

Authors: This is done.

Reviewer 3 Report

The authors have attempted to provide a review on estrogen receptor signaling pathways in breast cancer. The title implies that majority of the focus would be on estrogen receptor signaling in breast cancer however, majority of the article summarizes basic facts about breast cancer in general without adequate focus on ER+ subtype of breast cancer. The figures are redundant and do not add value to what has been known for over a decade. The text is poorly organized and less than 40% of the article details estrogen receptor signaling pathways.

Author Response

Reviewer 3:

The authors have attempted to provide a review on estrogen receptor signaling pathways in breast cancer. The title implies that majority of the focus would be on estrogen receptor signaling in breast cancer however, majority of the article summarizes basic facts about breast cancer in general without adequate focus on ER+ subtype of breast cancer. The figures are redundant and do not add value to what has been known for over a decade. The text is poorly organized and less than 40% of the article details estrogen receptor signaling pathways.

Authors: We have deeply reviewed the manuscript and reorganized it to make it clearer. We added different aspects of the genomic and non-genomic action of ER. Its interaction with Src tyrosine kinase and PI3K as well as the role of GPER in breast cancer. We also described the post-translational modifications of ERα and how these events influence both ligand-independent and non-genomic activity of ERα. We also mentioned the interactions between the nuclear and membrane-initiated pathways of ERα. The degree of detail in estrogen receptor signaling pathways has increased considerably now.

Round 2

Reviewer 1 Report

Authors addressed all my comments. The manuscript is suitable for publication.

Author Response

We have corrected the manuscript according to the last remark of the academic editor. 

Reviewer 2 Report

Thank you for the revision.

Minor

1.       Please make sure to correctly set spaces between words, these are often missing

Author Response

(The authors gave the same response as above.)

Reviewer 3 Report

There are several typos throughout the manuscript that need to be fixed. For example, the title for section 3-1 molecular mechanisms states ER3-1-1 which is incorrect. 

Author Response

(The authors gave the same response as above.)
